# A Label-Free Optical Flow Cytometry Based-Method for Rapid Assay of Disinfectants’ Bactericidal Activity

**DOI:** 10.3390/ijms25137158

**Published:** 2024-06-28

**Authors:** Andreea Maria Pîndaru, Luminița Măruțescu, Marcela Popa, Mariana Carmen Chifiriuc

**Affiliations:** 1Department of Botany and Microbiology, Faculty of Biology, University of Bucharest, 91-95 Splaiul Independentei, 050095 Bucharest, Romania; pindaru.andreea@s.bio.unibuc.ro (A.M.P.); carmen.chifiriuc@bio.unibuc.ro (M.C.C.); 2Research Institute of University of Bucharest, University of Bucharest, 050663 Bucharest, Romania; marcela.popa@bio.unibuc.ro

**Keywords:** label-free flow cytometry, quaternary ammonium compounds, disinfection

## Abstract

Selecting the appropriate disinfectant to control and prevent healthcare-associated infections (HAIs) is a challenging task for environmental health experts due to the large number of available disinfectant products. This study aimed to develop a label-free flow cytometry (FCM) method for the rapid evaluation of bactericidal activity and to compare its efficacy with that of standard qualitative/quantitative suspension tests. The bactericidal efficiency of eight commercial disinfectants containing quaternary ammonium compounds (QACs) was evaluated against four strains recommended by EN 13727 (*Escherichia coli*, *Pseudomonas aeruginosa*, *Staphylococcus aureus*, *Enterococcus hirae*) and four multidrug-resistant pathogens. The proposed FCM protocol measures changes in scattered light and counts following disinfectant exposure, neutralization, and culture steps. Unlike other available FCM-based methods, this approach does not rely on autofluorescence measurements, impedance cytometry, or fluorescent dyes. The FCM scattered light signals revealed both decreased count rates and morphological changes after treatment with minimum inhibitory concentrations (MICs) and higher concentrations for all tested bacteria. The results from the FCM measurements showed excellent correlation with those from standard assays, providing a rapid tool for monitoring the susceptibility profile of clinical, multidrug-resistant pathogens to chemical disinfectants, which could support infection prevention and control procedures for healthcare environments. This label-free FCM protocol offers a novel and rapid tool for environmental health experts, aiding in the optimization of disinfectant selection for the prevention and control of HAIs.

## 1. Introduction

Cleaning and disinfection are essential for controlling the spread of pathogens in healthcare settings. Daily antiseptic and disinfectant products based on alcohols, quaternary ammonium compounds (QACs), hydrogen peroxide or chlorine are intensively used to reduce the load of pathogens within hospital environments. Due to the COVID-19 epidemic, their use has increased not only in healthcare settings but also in homes, offices, and industrial environments. The QAC-based products were the most prevalent used in both nonalcoholic-based hand sanitizers and surface disinfectants for limiting virus spread [1,2,3]. At present, QACs products are used at a large scale, and it is estimated that these products will dominate the global surface disinfectants market, accounting for about 29% of the global market by 2030.

In vitro exposure of microorganisms from various environments (hospital, food processing, household, industry settings) to chemical disinfectants, including QAC-based products was demonstrated to result in reduced susceptibility [4,5,6,7,8,9]. For example, repeated sub-MIC/MIC exposure to QACs was demonstrated to induce mutations in regulators (*acrR*, *marR*, *soxR*, and *crp*), outer membrane proteins and transporters (mipA and sbmA), and RNA polymerase (*rpoB* and r*poC*) genes in *E. coli* [10]. Additionally, compared to other disinfectants those containing QACs as the sole active agent are more likely to select for resistance (sometimes combined with antibiotic resistance), as reported in pathogens imbedded in sink biofilms, thus contributing to outbreaks of healthcare-associated infections [1,11]. Therefore, it is crucial to assess the susceptibility profile of bacteria collected from environments with high usage of chemical disinfectants, such as clinical settings [1,12]. Testing these profiles allows for the detection of decreased susceptibility of pathogens to disinfectants, thereby enhancing infection prevention and control measures.

Currently, chemical disinfectants are evaluated with regard to their microbicidal activities using standard protocols based on cultivation methods (EN 13727:2012+A2:2015) [13]. These tests are based on exposure of planktonic (suspension tests) or adhered bacterial reference strains (carrier tests) to different disinfectant concentrations. After the indicated contact time, the effects of disinfectants are stopped by a neutralization step, and afterwards, the growth of bacteria is assessed by cultivation after 48 h. These cultivation methods are effective but laborious and time-consuming.

Optical flow cytometry (FCM) in combination with fluorescent labelling was demonstrated to be an effective and rapid method for antibiotic susceptibility testing. Also, it can be used for bacterial enumeration [14], viability and metabolic activity assessment, and the detection and identification of pathogens in different types of samples [15,16,17,18]. However, staining was shown to underestimate the number of dead cells. For example, the fluorescence of propidium iodide (PI) is affected by chlorine [19,20,21]. Other limitations of fluorescent labelling include cytotoxicity [22], underestimation of viability [23], species-specific responses [24], differences in binding affinity to live and dead cells [25], and autofluorescence of bacteria. Therefore, fluorescent labelling might not always be a reliable indicator for bacterial viability. Label-free optical FCM methods have been developed [17,26,27] (Table 1); however, these methods rely on detection of bacteria’s autofluorescence, not on their optical scattering proprieties. Additionally, not all bacteria exhibit autofluorescence.

An alternative to optical FCM is impedance FCM, which has proven to be competitive in several areas. In contrast to optical FCM, which relies on the detection of forward- and side-scattered light (FSC-A and SSC-A) to obtain information on cell size and cell granularity, impedance FCM records the cell electrical properties (Table 1). Label-free optical FCM or impedance cytometry can yield comparable data on the cells in suspension. However, the label-free nature, low acquisition costs, and high flexibility in configuration has resulted in several applications for impedance FCM. Examples include detection and enumeration of bacteria, viability assessment [28,29,30], and monitoring of compounds effects in drug exposure assays [31,32,33]. However, the main limitation of impedance FCM is that it does not allow the distinction of the different physiological states of bacteria (injured cells, active respiring cells, DNA damaged cells, depolarized cells, cells with enzymatic activity) that occur in response to disinfection procedures [34]. On the other hand, optical FCM with fluorescent labelling provides further information about the impact of antimicrobial compounds or physical treatments on microbial cells [34,35]. Understanding how various subpopulations behave upon exposure to antimicrobials is essential for their continued utilization and for prevention of resistance emergence.

Various bacteria use the VBNC state as a persistence mechanism to survive in different stress conditions encountered in the clinical and natural environments [36,37,38]. The bacteria in the VBNC state escape the conventional detection method and represent a reservoir for pathogenic bacteria which provides a continuous source for recurrent infections and transmission [39,40]. Therefore, novel strategies to detect viable pathogens that are not only based on plating assays are urgently required. Several studies have shown that optical FCM provides insight into damaged bacterial cell populations and thus can detect bacteria in the VBNC state, which may occur at subinhibitory concentrations of disinfectants [18,35,41,42,43,44].

The aim of this study was to develop a novel method based on optical FCM without fluorescent labelling for rapid evaluation of bactericidal activity of QAC-based disinfectants and evaluate its performance in comparison to conventional culture-based methods. The novelty of the present work was to use the changes in the light-scattering proprieties of bacteria exposed to different concentrations of QACs to directly determine disinfection efficacy. The proposed optical FCM method, which measures scattered light and count rates, is able to provide results within four hours. We further showed that optical FCM with LIVE/DEAD staining and CTC measurements of bacteria exposed to subinhibitory concentrations of QACs provides evidence that bacterial pathogens can enter the VBNC state. Our novel method could be used as an advanced monitoring tool of disinfection efficacy against clinically relevant pathogenic bacteria. It represents a powerful tool that may contribute to a better understanding of the persistence and spread of pathogens and thus to the protection of vulnerable hospitalized patients.

## 2. Results

### 2.1. Qualitative and Quantitative Suspension Test Results

To quantify the efficacy of the selected QAC disinfectants against reference and MDR bacterial strains, qualitative and quantitative suspensions tests were performed. The results of the two standard methods were overall similar. Table 2 summarizes the minimum bactericidal concentration (MBC) values recorded if a logarithmic reduction ≥5 in viable counts was achieved. Overall, the QAC disinfectants were efficient in inhibiting bacterial growth of all tested bacteria at concentrations lower than the label concentrations recommended by the manufacturer. The lowest MBC values were detected for Dezicon^®^, with values between 0.0015% and 0.001%. Compared with all other strains, significantly higher MBC values were determined for Sterisol^®^, Desogen^®^, Dezicon^®^, Terralin Protect^®^, and Cleanisept^®^ against the multidrug-resistant *P. aeruginosa* 1707. The MBC values determined for Desogen^®^, Cleanisept^®^, Terralin Protect^®^, Dezicon^®^, Isorapid^®^, and Sterisol^®^ against Gram-positive bacteria were lower than those obtained for Gram-negative bacteria. Compared to all the tested disinfectants, Sterisol^®^ was the least active, with MBC values ranging from 5% to 0.039%.

### 2.2. Rapid Enumeration of Potentially Viable Bacteria Using Label-Free FCM

Using the optical FCM protocol as described, we tested standardized bacterial suspensions of the tested bacterial strains. Cell numbers were set at approximatively 10^6^ CFU/mL in filtered PBS. Agreement was found within approximatively 0.5 log_10_ CFU/mL (Figure 1). The results showed that the FCM counting slightly overestimated the cell numbers by comparison with a standard assay based on cultivation.

### 2.3. Altered FCM Scatter Patterns Are Associated with Disinfectant Activity

To test the application of FCM for the evaluation of bactericidal activity of QAC disinfectants, we selected eight bacterial strains: four reference strains and four MDR clinical strains. The principle of disinfectant efficiency assessment using the optical FCM method is that following disinfectant exposure, neutralization and culture steps (incubation for 4 h at 35 ± 2 °C), the bacterial cells will exhibit both altered biology and counts that can be measured and converted into prediction of MBC values.

The data obtained from the untreated bacteria (one well of untreated bacteria per disinfectant on the microtiter plate, measured before the disinfectant-treated bacteria) were used to define the “Untreated bacteria gate” in Kaluza^®^ analysis software (Beckman Coulter, Brea, CA, USA) (Figure 2). Further, the “Untreated bacteria gate” was passed down to all disinfectants’ treated samples.

The results showed that at concentrations above and near MBC values, the disinfectant-treated samples exhibited alterations in both the scatter patterns and counts as compared to the untreated samples. By contrast, for samples treated at subinhibitory concentrations, the optical FCM measurements indicated light-scatter signatures similar to the untreated bacteria (Figure 3). Further, the bacterial counts/µL determined for each disinfectant concentration were adjusted to the negative and positive controls, resulting in modified bacterial counts/µL. The results corresponding to disinfectants’ MBC and 1/2 MBC values are presented in Figure 4. At concentrations above and near disinfectants’ MBC values, we found that the optical FCM counts/μL falling in the region “Untreated bacteria” were drastically decreased compared to those obtained for subinhibitory concentrations (1/2 MBC values) (*p* < 0.0001) and untreated bacteria (*p* < 0.0001).

We performed point-to-point comparisons of the FCM data (counts/μL) obtained for the disinfectant-treated samples with standard culture-based assays results. The FCM results paired with the standard assays’ results were then converted to binary results “above or MBC” or “not MBC”. Using the ROC curve analysis, we determined the empirical cut-off of <0.1 bacteria-like particles counts/μL for disinfectant bactericidal effects. The area under the ROC curve was 0.96, standard error 0.012, 95% confidence interval, *p* < 0.001. This yielded an 89.8% sensitivity, 99.2% specificity, for confirmation of disinfectants’ MBC values (Figure 5). Based on the cut-off value of <0.1, agreement between the novel FCM protocol results and standard culture-based assay, i.e., whether both assays detected “above or MBC values”, was 147 (87.5%) over 168 individual assay results. Numerical comparisons of MBC by FCM and standard suspension tests showed that 96.42% (162/168) of all results were within ±1 doubling dilution (Table 3, Figure 6).

### 2.4. Viability State of Bacteria following Exposure to QAC Disinfectants

FCM with staining was carried out for all disinfectant-treated samples at MBC and subinhibitory concentrations and for untreated controls. The viability tests were performed using two commercial kits: BacLight^™^ Redox Sensor^™^ Green Vitality Kit and LIVE/DEAD^™^ BacLight^™^ Bacterial Viability. Using fluorescent-labelled untreated (viable) and dead (heat-treated) controls, the “viable cells” gate, “dead cells” gate, and “active respiring cells” gate, respectively, were defined. Subsequently, the viability of the bacteria exposed to disinfectant concentrations lower than MBC (i.e., 1/2 MBC) was evaluated. The FCM results obtained for all the tested bacteria treated at disinfectant subinhibitory concentrations indicated viable and metabolically active cell populations. Figure 7 depicts, as demonstrative examples, the FCM profiles obtained for two test strains, *E. hirae* ATCC 10541 (Gram-positive) and *E. coli* K12NCTC 10538 (Gram-negative), exposed to subinhibitory concentrations of QAC-based disinfectants, Incidin Pro^®^ and Sterisol^®^. Apart from the dead and viable cell populations, a distinct population of injured cells was noted for the tested Gram-negative bacteria exposed to subinhibitory concentrations of Desogen^®^, Dezicon^®^, Terralin Protect^®^, Isorapid^®^, Incidin Pro^®^, and Cleanisept^®^.

The FCM counts/μL for samples treated with disinfectant at MBC and 1/2 MBC concentrations were determined for each QAC-based disinfectant and then adjusted to the negative and positive controls. The results are presented in Figure 8. At MBC values, we found that the FCM counts/μL falling in the regions “viable” and “active respiring” were drastically decreased compared to those obtained for samples treated with subinhibitory concentrations (1/2 MBC values) (*p* < 0.0001) (Figure 8).

## 3. Discussion

Currently, evaluation of chemical disinfectant efficacy is performed using standard assays which rely on bacteria cultivation after disinfectant exposure. To overcome the limitations of standard assays, including that they are laborious to perform and take a long time until results are obtained, we investigated the potential application of a label-free optical FCM-based method for rapid assessment of QAC disinfectant bactericidal activity. The developed optical label-free FCM protocol was applied to a total of eight QAC disinfectants used for surface disinfection in the medical field and evaluated on reference and MDR bacteria belonging to the ESKAPEE group, which are able to persist and pose a serious risk of infections in hospital facilities [29]. We focused our technical demonstration on QAC disinfectants for the following reasons: they are used extensively in healthcare facilities [1,11], subinhibitory concentrations can lead to QACs-tolerant phenotypes [2], and QAC tolerance genes are shared by diverse bacterial taxa within the same ecosystem and are often associated with antibiotic resistance genes [2,45].

Differences among bacteria tested with regard to their susceptibility to QAC disinfectants were noticed. Significantly higher (8-fold) bactericidal concentrations for two of the tested QAC disinfectants, Sterisol^™^ and Terralin Protect^™^, were observed for the *P. aeruginosa* clinical strain, highlighting the need for disinfectant testing methods that consider the strain variances and the necessity for monitoring a pathogen’s susceptibility to disinfectants. In this context, one limitation of our study is the limited range of organisms tested, underscoring the need for future research to expand the scope.

Our results clearly demonstrate that the results of the label-free optical FCM-based method applied to bacteria exposed to QAC disinfectants were well corroborated with those obtained from standard suspension tests. The main advantage of the optical FCM approach is the fast analysis results (within 4 h) compared to the standard methods (48 h). The FCM light-scatter signatures of disinfectant-treated bacterial cells were correlated with the bactericidal effects of QAC disinfectants. Based on both altered light-scatter profiles and the drastic decrease in count rates, the label-free optical FCM method was able to stratify the disinfectant-treated samples as “above or MBC” and “not MBC”. The developed optical FCM method has an 89.8% sensitivity and 99.2% specificity for determining QAC disinfectants’ bactericidal activity. The great majority (96.85%) of the optical FCM results were within ±1 doubling dilution with the standard suspension tests. Discrepancies between the standard method and optical FCM protocol were detected in the case of the *P. aeruginosa* clinical strain and *S. aureus* ATCC 6538 treated with Desogen^™^. In these cases, the disinfectant-treated samples classified as “not MBC” by standard suspension tests were classified as “MBC” by the FCM method. This might be explained by the fact that glutaraldehyde (active ingredient of Desogen^™^) affects DNA, RNA, and protein synthesis [46], which results in slower bacterial growth for these strains, which was not detected by the optical FCM method.

The developed label-free optical FCM method is advantageous in outbreak situations when environmental health experts need to act quickly and select effective disinfectants. For example, *Mycobacterium massiliense* was found in 38 hospitals in Rio de Janeiro (Brazil) during August 2006–July 2007 [47]. Although the origin of the outbreak was unknown, the strains that caused it were not only clinically resistant to ciprofloxacin, cefoxitin, and doxycycline but also resistant to glutaraldehyde (2% *w*/*v*), which was employed at the time to disinfect endoscopes. It would have been important at the time to have access to a rapid tool for the evaluation of disinfectants’ bactericidal activity to limit the spread of infection.

Injured bacterial populations were detected using optical FCM in combination with fluorescent dyes for *E. coli* and *P. aeruginosa* strains exposed to subinhibitory concentrations of Desogen^®^, Dezicon^®^, Terralin Protect^®^, Isorapid^®^, Incidin Pro^®^, and Cleanisept^®^. These results suggest the increased persistence of *E. coli* and *P. aeruginosa* in a sublethal state and their potential role in HAIs linked to the application of inappropriate disinfectant concentrations. Therefore, research in this area should be undertaken.

In this work we demonstrated that by using optical FCM, it is possible to evaluate changes in both scattered light and count rates following disinfectant exposure and that the method is easy to resolve without fluorescent labelling. Our method may face a few challenges regarding its introduction into clinical settings. Although personnel with the necessary competence in clinical microbiology laboratories can prepare samples and operate flow cytometers, training is required to interpret optical FCM results. A significant challenge is the need for automation of optical FCM data processing. However, an automated FCM approach for water chlorination efficacy assessment was developed [48]. Another challenge is represented by the fact that the method’s performance across all types of pathogens causing HAIs is unknown. Despite not being detected in this study, discrepancies with nucleic acid staining phenotypes across different species and settings emphasize the need for further research [18].

## 4. Materials and Methods

### 4.1. Bacterial Strains and QAC-Based Disinfectants Tested

Eight bacterial strains were used for bactericidal efficiency tests, represented by four reference strains, *Staphylococcus aureus* ATCC 6538, *Pseudomonas aeruginosa* ATCC 15442, *Enterococcus hirae* ATCC 10541, and *Escherichia coli* K12 NCTC 10538, and four multidrug-resistant (MDR) clinical strains (Table 4). Frozen bacterial stocks were inoculated on Plate Count Agar (PCA) (Scharlau, Barcelona, Spain) and incubated, under aerobic conditions, at 35 ± 2 °C, for 18–24 h. The precultures were further inoculated on Tryptone Soy Agar (TSA) (Scharlau, Barcelona, Spain) and grown overnight aerobically at 35 ± 2 °C. Subsequently, bacterial test suspensions with a cell density of (~5 × 10^8^ cells)/mL were prepared in sterile Tryptone buffer and used for bacterial efficacy assays as described in the European Standard EN 13727+A2 [13], albeit with modifications to adapt the technique to a microscale level.

Eight QACs containing products of commercial grade used for disinfection and cleaning of medical devices and surfaces in the hospital environment were selected. Their characteristics are listed in Table 5. The QAC-based disinfectants were purchased from local distributors. Serial two-fold dilutions were prepared immediately before testing.

### 4.2. Qualitative and Quantitative Suspension Tests

Disinfectant testing was performed according to the guidelines of the EN 13727+A2 [13] Antiseptics and chemical disinfectants—Quantitative suspension testing for the evaluation of bactericidal activity in the medical field, test method and requirements (phase 2, stage 1) [12]. Bactericidal concentrations were determined by qualitative and quantitative suspension tests, albeit adapted to the microscale level using a 96-well microtiter plate as described previously [18,19].

Bactericidal concentrations were determined by qualitative and quantitative suspension tests. A volume of 20 μL of bacterial test suspension was added to each well containing 160 μL of disinfectant dilution and 20 μL of water and gently mixed. After the respective exposure times recommended by the manufacturer (Table 4), at approximatively 20–22 °C (room temperature), 20 μL aliquots were transferred to 160 μL of Neutralizing Fluid (Scharlau, Eur. Pharm., Barcelona, Spain) for 5 min. Subsequently, bacterial test suspensions were inoculated in Tryptone Soy Broth (TSB) (1:10 in TSB). Bacterial growth was assessed visually after 24 h of incubation at 35 ± 2 °C. For quantitative suspensions tests, at the end of neutralization time, the bacterial test suspensions were plated on TSA for 48 h at 37 °C. Logarithmic reductions (LRs) in viable counts were calculated based on bacterial counts. An LR ≥ 5 in viable counts was required for bactericidal activity. Untreated growth controls were included for all assays. The tests were performed in duplicate and repeated three times.

### 4.3. Flow Cytometer Set-up

All the assays were performed using a BD AccuriC6 Plus^™^ (BD Bioscience, San Jose, CA, USA) equipped with a 488 nm blue laser (FL1 533/30 nm, FL2 585/40 nm, FL3 > 670 nm) and 640 nm red laser (FL4 675/25 nm). Instrument calibration was performed with BD^®^ CS&T performance tracking beads (cat 661414). The instrument settings for counting were as follows: unstained—FSC 1000 (threshold) and SSC 1000. The acquisition rate was set at 35 µL per minute by 10-fold serial dilution and acquired for ≥25 µL. For fluorescence detection, the FL1 (533/30) for Syto^™^24 or Syto^™^9 and FL3 (670LP) detectors for PI or CTC were used. Trigger was set at 5000 on FL1 (Syto^™^24 or Syto^™^9). Voltages were optimized for each detector, and compensation was performed for PI/SYTO^™^9 according to the manufacturer’s recommendations. Up to 5000 cells were analyzed per sample. FCM data were analyzed with AccuriC6 Plus^®^ software or Kaluza^®^ analysis software 2.2 (Beckman Coulter, Brea, CA, USA).

### 4.4. Gating and Counting of Bacteria by Label-Free FCM and Plating

Determination of bacterial counts using the label-free optical FCM method and conventional plating was performed using the eight bacterial strains. Suspensions were prepared in filtered–sterilized phosphate buffered saline (PBS) from fresh bacterial cultures (16–18 h). Each bacterial suspension was standardized using the 0.5 McFarland turbidity standard to a density of approximatively 1.0 × 10^8^ CFU/mL and then further diluted to 10^6^ CFU/mL in filtered PBS. The unstained bacterial suspensions were analyzed by FCM and by conventional plating. Ten-fold serial dilutions of the suspensions were prepared in PBS and cultured on TSA plates. After incubation at 35 ± 2 °C for 24 h, the bacterial counts were recorded.

The same suspensions of 10^6^ CFU/mL in PBS and 10-fold serial dilutions were acquired using a BD AccuriC6 Plus^™^ with the set-up described as above. A conventional FCM instrument is able to detect and count bacteria as single particles [31,45] since the size of most bacteria varies between 500 and 1000 nm. Distinguishing populations of cells can be relatively straightforward for samples containing only one type of cell. Hence. bacteria were identified based on the forward-scatter (FSC) and side-scatter (SSC) signals on the log scale (scatter patterns). A gate for bacterial population was set for each tested suspension. The events corresponding the bacterial population were assessed only in the bacteria gate. The scatter events outside this gate were excluded from quantitative analysis. The assays were performed in triplicate.

### 4.5. Bacterial Growth Evaluation Using FCM without Labelling

To assess the bacterial growth, the disinfectant-treated bacteria and controls (untreated) following neutralization were inoculated 1:10 in media culture (TSB). After 4 h of incubation at 35 ± 2 °C, the samples were washed with filtered PBS. For each assay, the FSC and SSC signals of the untreated control bacteria were used to distinguish bacterial cells from instrumental noise and to set the gate on FSC/SSC log scale, corresponding to morphological characteristics of the bacteria (scatter patterns). The untreated (control bacterial populations) and disinfectant-treated bacteria were counted only in the gate corresponding to the bacterial population. The scatter events outside the bacteria control gate were excluded from quantitative analysis. For enumeration of bacteria, counting beads were used that were detected by FSC and SSC parameters. The bacterial counts corresponding to each disinfectant dilution were adjusted by scaling them to their respective negative control (no bacteria) and positive control (growth control), resulting in modified bacterial counts [49]. Each assay was performed in duplicate and repeated three times.

### 4.6. Bacterial Viability Evaluation Using FCM with Fluorescent Labelling

For all samples, the bacterial viability was determined using FCM with fluorescent labelling. LIVE/DEAD^®^ Bac Light^™^ (Thermo Fisher Scientific, Waltham, MA, USA) and Bac Light^®^ Redox Sensor^™^ CTC Vitality (Thermo Fisher Scientific, USA) kits were used separately for viability evaluation. The staining protocols followed the manufacturer recommendations. Each acquisition included an unstained sample to define the positive staining gate and single stain controls.

The LIVE/DEAD^®^ Bac Light^™^ kit allowed the detection of viable cells based on their membrane integrity. Cells with altered membrane were considered to be not viable or dying and were stained with propidium iodide (PI) in red, whereas cells with an intact membrane were stained with Syto9 in green. For staining, 1.5 μL of PI (20 nM in DMSO) and 1.5 μL of SYTO^®^9 (3.34 mM in DMSO) were added to each sample, diluted 10-fold in saline, and incubated for 15 min in the dark before FCM analysis. The Bac Light^®^ Redox Sensor CTC Vitality Kit allowed the detection of active respiring cells (metabolic active cells). Syto^®^24 (final concentration 1:10,000) was used for total cell counting, whereas 5-cyano-2, 3-ditolyl tetrazolium (CTC) (final concentration 2 mM, 1 h at 35 ± 2 °C) was used for assessment of active respiring cells. Staining procedures were all conducted in the dark.

### 4.7. Flow Cytometry Gating Strategy

Flow cytometry standard (FCS) files exported for each sample were analyzed using Kaluza^®^ software 2.2 (Beckman Coulter). The bacterial population was distinguished using an FSC-H versus SSC-H plot (log scale), and most events were gated to be included. Using a plot SSC area versus SSC-H, doublet discrimination was performed. Subset gating was the final step in the gating process.

LIVE/DEAD^®^ Bac Light^™^ gating was performed as instructed by the manufacturer with viable (intact membrane, Syto^™^9 fluorescence), dead (totally permeabilized membrane, PI fluorescence), and injured (damaged membranes, Syto^™^9/PI fluorescence) cells. The gating for Bac Light^®^ Redox Sensor^™^ CTC Vitality included total (acid nucleic stain, Syto^™^9 fluorescence) and active respiring (CTC fluorescence) cells, as recommended by the manufacturer. Gates for viable cells were drawn based on the events in the untreated control samples containing viable culture only, while gates for dead cells were based on samples containing fixed/dead cells only and adjusted slightly as needed. LIVE/DEAD^®^ Bac Light^™^ gating included viable (intact membrane, Syto^™^9 fluorescence), dead (totally permeabilized membrane, PI fluorescence), and injured (damaged membranes, Syto^™^9 and PI fluorescence) cells. BacLight^®^ Redox Sensor^™^ CTC Vitality included total (acid nucleic stain, Syto^™^9 fluorescence) and active respiring (CTC fluorescence) cells. Using the gates drawn, the counts/μL in each gate (“viable”, “dead”, “total cells”, “active respiring cells”) were calculated.

### 4.8. Statistical Analysis

GraphPad Prism 5 (GraphPad Software, San Diego, CA, USA) was used to performed the statistical tests. Difference in cell-event enumeration between samples (untreated and disinfectant-treated samples at different concentrations) were assessed using a Kruskal–Wallis nonparametric one-way ANOVA with Dunn’s multiple comparisons.

## 5. Conclusions

The developed label-free optical FCM method provided rapid and quantitative analysis of the bactericidal activity of QAC-based disinfectants, demonstrating an excellent correlation of over 96.42% with conventional suspension tests. This label-free optical FCM protocol offers a valuable tool for environmental health experts to optimize the selection of disinfectants for preventing and controlling HAIs. Additionally, the optical FCM method enhances our understanding of the phenotypic response following biocidal product interaction with bacterial cells. It also provides important insights into injured bacterial cell populations, particularly when subinhibitory concentrations of disinfectant products are used.

## Figures and Tables

**Figure 1 ijms-25-07158-f001:**
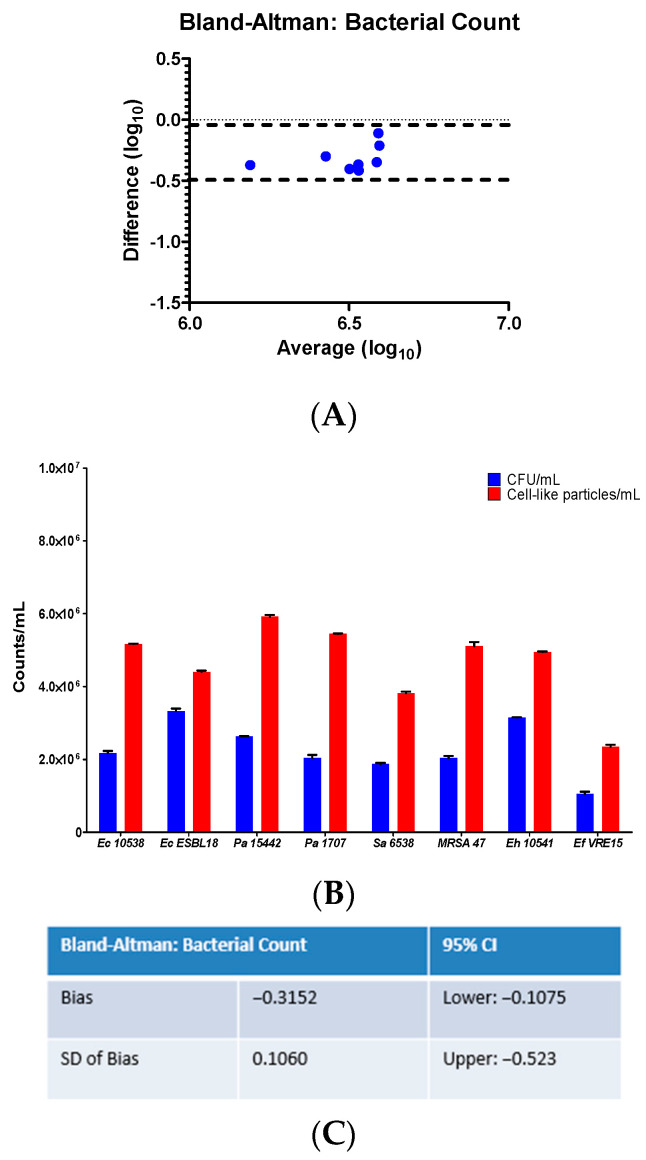
Enumerating bacteria using label-free FCM and plating. (**A**,**B**) The quantitative analysis of bacterial cells numbers set at approximatively 10^6^ CFU/mL indicated that FCM overestimated the bacterial numbers in comparison with plating. (**C**) Agreement was found within approximately 0.5 log_10_ CFU/mL. Ec (*E. coli*), Pa (*P. aeruginosa*), Sa (*S. aureus*), MRSA (methicillin-resistant *S. aureus*), Eh (*E. hirae*), Ef (*E. faecium*).

**Figure 2 ijms-25-07158-f002:**
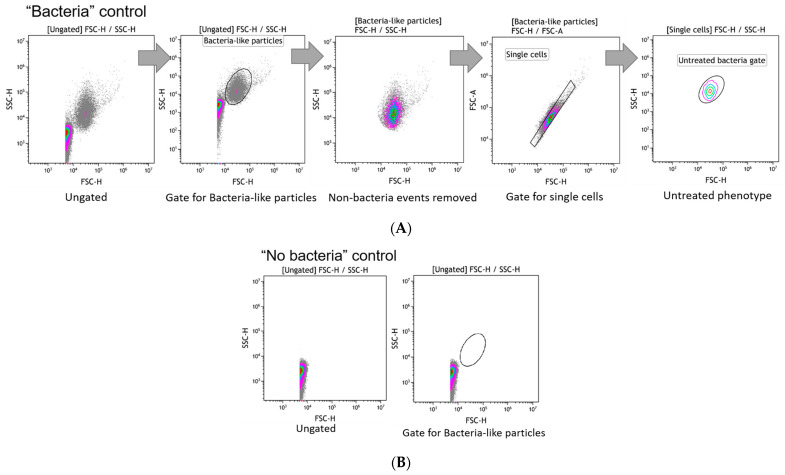
FCM gating workflow for bacteria-like particles counting. (**A**) An unstained aliquot of the standardized bacterial suspension (“bacteria” control) is measured and the gate for bacterial-like particles is set based on the FSC/SSC signals on log scale. The sample is back-gated to verify that non-bacteria events were removed. A bivariate plot (FSC-A vs. FSC-H) is used to exclude aggregates and select further only single cells. Then bacteria-like events are mapped with a 10% nearest neighbour contour visualization and a gate is set. This gate is named as the Untreated phenotype and is passed down to all the disinfectant-treated samples in the assay. (**B**) “No bacteria” control plot shows that no events are detected within the bacteria-like particles gate.

**Figure 3 ijms-25-07158-f003:**
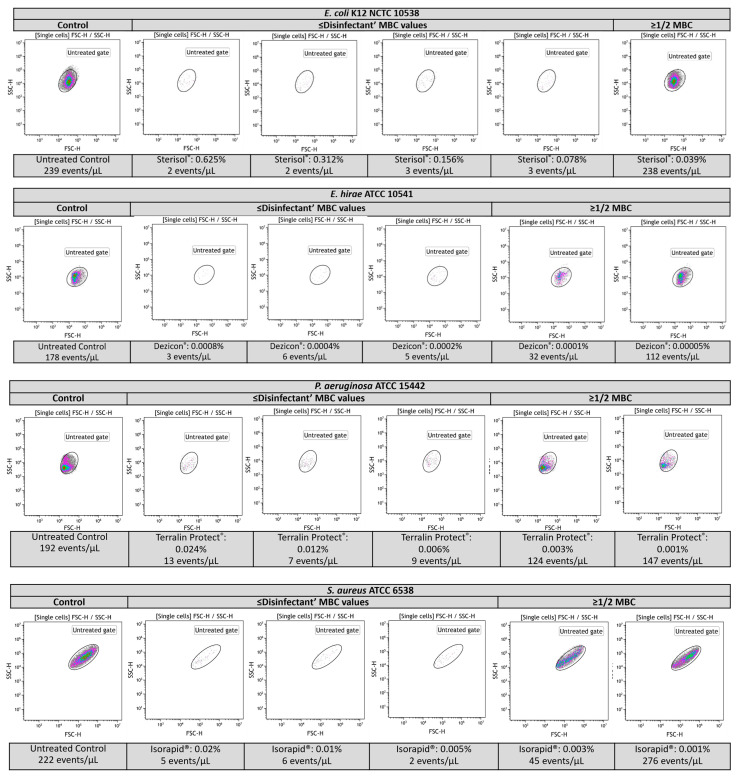
FCM light-scatter patterns are associated with disinfectants’ bactericidal effects. Results analysis of the FCM signals obtained for the untreated control bacteria and disinfectant (Sterisol^®^, Dezicon^®^, Terralin Protect^®^, Isorapid^®^)-treated bacteria. The results showed changes in scatter profiles and counts correlated with antimicrobial activity at concentrations above and near MBC values. These altered scatter patterns and counts were detected for all the bacteria and QAC disinfectants tested. *E. coli* K12 NCTC 10538, *E. hirae* ATCC 10541, *P. aeruginosa* ATCC 15442 and *S. aureus* ATCC 6538 are presented as demonstrative examples.

**Figure 4 ijms-25-07158-f004:**
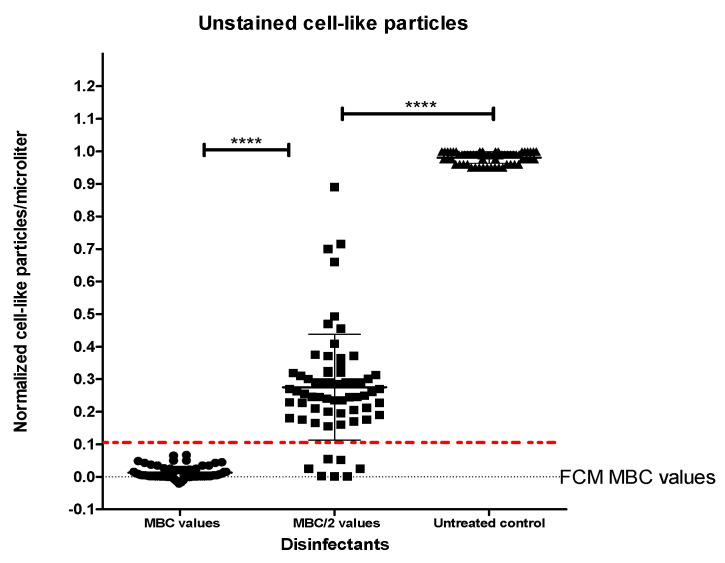
Bacterial counts/µL are associated with disinfectant antimicrobial activity. Significant differences were detected between samples treated at MBC values, 1/2 MBC, and untreated (**** *p* value < 0.0001). The dashed line indicates the empirical cut-off of <0.1 bacteria-like particles counts/μL for disinfectant bactericidal effects.

**Figure 5 ijms-25-07158-f005:**
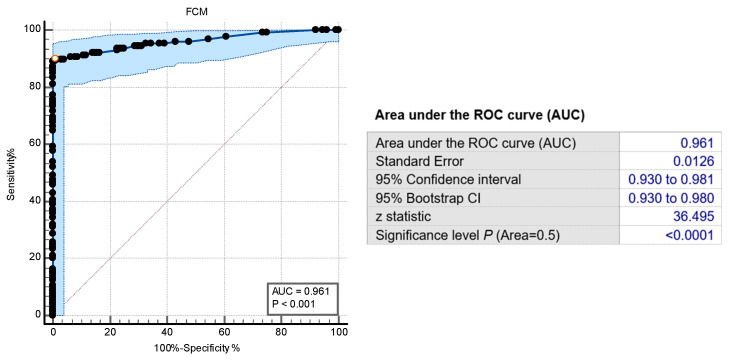
Modified bacterial counts/μL predict the disinfectants’ MBC values. The new FCM protocol showed excellent prediction of disinfectants’ MBC values when a cut-off of <0.1 modified bacterial counts/μL was applied (sensitivity 0.89, specificity 0.99).

**Figure 6 ijms-25-07158-f006:**
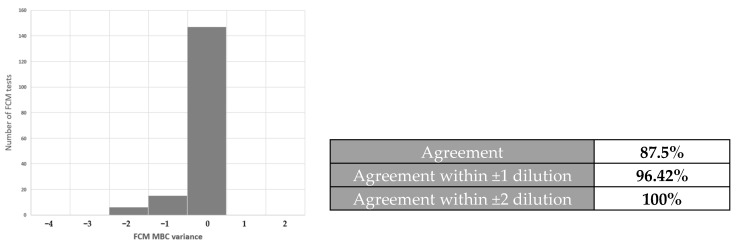
Agreement of FCM method with standard suspension tests. A total of eight bacterial strains were tested against eight QAC disinfectants. A total of 147 out of 168 FCM assays results were in agreement (87.5%) with the standard method. A total of 96.85% FCM results were within ±1 doubling dilution with the standard suspension tests.

**Figure 7 ijms-25-07158-f007:**
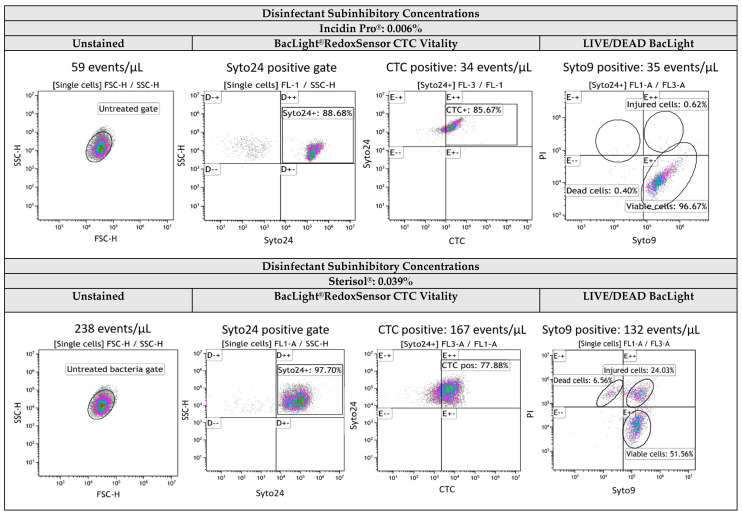
Flow cytometric density plots of *E. hirae* ATCC 10541 and *E. coli* K12 NCTC 10538 treated with chemical disinfectants Incidin Pro^®^ and Sterisol^®^, respectively, at subinhibitory concentrations and stained with BacLight^®^RedoxSensor CTC Vitality kit and LIVE/DEAD BacLight bacterial viability kit, respectively. FL1-A (BP 533/30 collecting SYTO^®^9 fluorescence) is plotted against FL3-A (LP ≥ 670 collecting PI fluorescence). The fluorescent staining with either Syto^®^24 and CTC or Syto^®^9 and PI demonstrated that the population of cell-like particles detected for samples treated with disinfectant subinhibitory concentrations were viable and active respiring bacteria.

**Figure 8 ijms-25-07158-f008:**
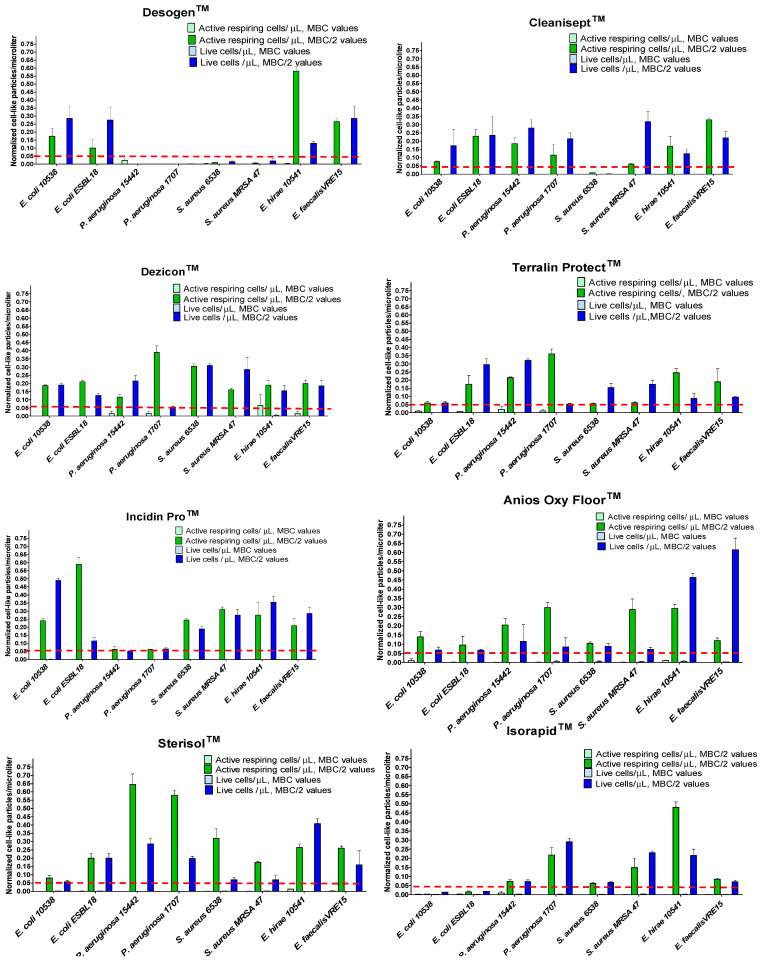
Modified bacterial counts/μL corresponding to viable and active respiring cell populations, determined for bacteria treated with each QAC disinfectant at MBC and 1/2 MBC values after 4 h incubation. The results showed that for samples treated at disinfectant MBC values, the FCM counts/μL falling in the regions “viable” and “active respiring” were drastically decreased (the dashed red line indicates counts/µL < 0.05), compared to those obtained for samples treated with subinhibitory concentrations (MBC/2 values) (*p* < 0.0001).

**Table 1 ijms-25-07158-t001:** The novelty of our developed label-free optical FCM method in comparison with previous label-free FCM methods.

References	Current Label-Free FCM Methods	Novelty of Our Developed Label-Free Optical FCM Method
[26,27]	Label-free optical FCM methods based on measurement of bacteria autofluorescence.	Our label-free FCM method uses light-scatter signal measurements for fast and accurate quantification of viable bacteria after exposure to disinfectants. It is not based on autofluorescence measurements.
[28,29,30,31]	Impedance FCM methods based on the measurement of cell electrical proprieties. The impedance FCM does not allow the differentiation of various physiological states of bacteria.	Our label-free optical FCM method detects and quantifies bacteria based on forward- and side-scattered light (FSC-H and SSC-H) measurements. Further, the optical FCM with fluorescent labelling provides information about the impact of antimicrobial agents on microbial cells.

**Table 2 ijms-25-07158-t002:** Bactericidal concentrations (%) resulting in a logarithmic reduction ≥5 determined by quantitative suspension tests.

Bacterial Strains Tested	Bactericidal Concentrations (%) of QAC Disinfectants							
	Sterisol^®^	Desogen^®^	Dezicon^®^	Terralin Protect^®^	Anios Oxyflor^®^	Isorapid^®^	Incidin Pro^®^	Cleanisept^®^
*E. coli* K12 NCTC 10538	0.078	0.006	0.0004	0.001	0.05	0.005	0.006	0.012
*E. coli* ESBL17	0.156	0.002	0.0004	0.001	0.003	0.097	0.003	0.025
*P. aeruginosa* ATCC 15442	0.625	0.05	0.0008	0.006	0.003	0.097	0.006	0.2
*P. aeruginosa* 1707	5	0.05	0.0015	0.05	0.006	0.019	0.003	0.05
*S. aureus* ATCC 6538	0.078	0.006	0.0002	0.006	0.025	0.005	0.012	0.006
*S. aureus* MRSA47	0.078	0.006	0.0004	0.006	0.003	0.003	0.003	0.006
*E. hirae* ATCC 10541	0.039	0.006	0.0005	0.006	0.05	0.001	0.003	0.012
*E. faecium* VRE 15	0.039	0.006	0.0001	0.003	0.006	0.002	0.012	0.003

**Table 3 ijms-25-07158-t003:** Agreement between FCM method and the standard suspension tests.

Disinfectants		Bacterial Strains										
		*E. coli* K12 NCTC 10538	*E. coli* ESBL17	*P. aeruginosa* ATCC 15442	*P. aeruginosa* 1707	*S. aureus* ATCC 6538	*S. aureus* MRSA47	*E. hirae* ATCC 10541	*E. faecium* VRE 15	Agreement	Agreement within ±1 Dilution	Agreement within ±2 Dilution
**Sterisol^®^**										8/8 (100%)		
**Desogen^®^**										4/8 (50%)	6/8 (75%)	8/8 (100%)
**Dezicon^®^**										8/8 (100%)		
**Terralin Protect^®^**										8/8 (100%)		
**Anios Oxyflor^®^**										8/8 (100%)		
**Isorapid^®^**										6/8 (75%)	8/8 (100%)	
**Incidin Pro^®^**										8/8 (100%)		
**Cleanisept^®^**										7/8 (87.5%)	8/8 (100%)	
	Agreement											
	Error is within ±1 dilution											
	Error is within ±2 dilution											

**Table 4 ijms-25-07158-t004:** Characteristics of the bacteria used for bactericidal efficiency tests.

Bacterial Strains	Source	Resistance Phenotype	Antibiotic Resistance Patterns
*S. aureus* ATCC 6538	Reference strain	-	
*P. aeruginosa* ATCC 15442	Reference strain	-	
*E. hirae* ATCC 10541	Reference strain	-	
*E. coli* K12 NCTC 10538	Reference strain	-	
*S. aureus* MRSA47	ND	MRSA	FOX, AZM, P, CN, E, TE, LZD
*P. aeruginosa* 1707	Tracheal secretion	CRPA	CAZ, A^™^, FEP, MEM, IMP, AK, TOB, CIP, CN, DOR
*E. faecalis* VRE 15	Tracheal secretion	VRE	TE, CN, CIP, P, VA
*E. coli* ESBL17	Urine	ESBL	CZ, PRL, AMP, CTX, A^™^, CXM, FEP, AMC, TE, CIP, SXT

Legend: AMC, amoxicillin–clavulanic acid; AK, amikacin; A^™^, aztreonam; AZM, azithromycin; CAZ, ceftazidime; CIP, ciprofloxacin; CN, gentamycin; CRPA, carbapenem-resistant *P. aeruginosa*; CTX, cefotaxime; CZ, cephazolin; CXM, cefuroxime; DOR, doripenem; E, erytomycin; ESBL, extended beta-lactamase; FOX, cefoxitin; FEP, cefepime; IMP, imipenem; LZD, linezolid; MEM, meropenem; MRSA, methicillin-resistant *S. aureus*; ND, not determined; P, penicillin; PRL, piperacillin; SXT, sulfamethoxazole–trimethoprim; TE, tetracycline; TOB, tobramycin; VA, vancomycin; VRE, vancomycin-resistant *Enterococcus*.

**Table 5 ijms-25-07158-t005:** Disinfectant products, active ingredients, and label conditions.

Disinfectant Products Containing QACs	Active Agents	Concentration Tested	Contact Time
Sterisol^®^ *4th generation*	Didecyldimethylammonium chloride, concentration in metric units 0.34%. Quaternary ammonium compounds, benzyl-C12-18-alkyldimethyl, chlorides, concentration in metric units 0.09%.	RTU	5 min
Desogen^®^ *3rd generation*	10% glutaraldehyde. 15% benzyl-C12-C18-alkyldimethyl, chlorides.	1%	5 min
Dezicon^®^ *4th generation*	17% di decyl dimethyl ammonium chloride. 4.5% benzyl C12-C18-alkyl dimethyl chlorides.	RTU	5 min
Terralin Protect^®^ *3rd generation*	22% benzyl-C12-18-alkyldimethylammonium chloride. 17% 2-phenoxyethanol. 0.9% amino alkyl glycine.	2%	15 min
Anios Oxy Floor^®^ *3rd generation*	750 ppm peracetic acid. 0.012% *N*-alkyl (C12-14)-*N*-benzyl-*N*, *N*-dimethyl ammonium chloride.	0.5%	5 min
Isorapid^®^ *5th generation*	20% ethanol. 28% 1-propanol. 0.056% quaternary ammonium compounds.	RTU	1 min
Incidin Pro^™^ *3rd generation*	10% 2-phenoxyethanol. 8% *N*,*N*-bis-(3-aminopropyl) dodecyl amine. 7.5% benzalkonium chloride.	0.25%	5 min
Cleanisept^™^ *5th generation*	3.33% di decyl dimethyl ammonium chloride. 6.66% quaternary ammonium compounds, benzyl-C12-16-alkyldimethyl-chlorides.	0.25%	5 min

Legend: ppm, parts per million; RTU, ready-to-use.

## Data Availability

Data are contained within the article.

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
