# Peer review of "A Label-Free Optical Flow Cytometry Based-Method for Rapid Assay of Disinfectants’ Bactericidal Activity"

_ijms, 2024, doi:10.3390/ijms25137158_

Round 1

Reviewer 1 Report (Previous Reviewer 1)

Comments and Suggestions for Authors

The authors have made very few changes to the manuscript to improve its quality, this is clear by comparing with the previous versions. Additionally, my original opinion still stands: label-free cytometry for bacteria assessment is not novel, as already published in these studies that use the bacteria’s autofluorescence:

https://pubs.acs.org/doi/10.1021/acs.analchem.9b01869

https://www.ncbi.nlm.nih.gov/pmc/articles/PMC3754729/

https://europepmc.org/article/med/22243282

Moreover, other label-free flow cytometry methods for bacteria detection have also been developed already:

https://www.mdpi.com/1424-8220/18/10/3496

https://www.researchgate.net/publication/374829508_Label-free_multidimensional_bacterial_characterization_with_ultrawide_detectable_concentration_by_microfluidic_impedance_cytometry

Thus, the novelty of the developed method is still lacking.

The authors seem to have resubmitted the manuscript with very little changes, and no new experiments and/or argumentation/discussion in order to better support the manuscript’s publication in such a high level Q1 journal such as IJMS.

Author Response

Reviewer 2 Report (New Reviewer)

Comments and Suggestions for Authors

The article “Label-free flow cytometry method for rapid assay of disinfectants’ bactericidal activity” submitted to the MDPI journal international journal of molecular sciences describes the development of a novel label-free flow cytometric assay to analyse disinfectants ability to kill bacteria, based on FSC and SSC alone. The method is novel, as highlighted by the authors, however there are major and minor changes that need to be incorporated before the manuscript can be accepted for publication.

Major Comments:

1.      Whilst the authors have implemented additional references in response to a previous reviewer, the final paragraph of the introduction should emphasise the novelty of the developed assay, summarising what the authors presented in the table (in the response letter to previous reviewer).

2.      The Table 1 legend should have abbreviations listed in alphabetical order – the order is currently non-sensical.

3.      The table legends should include ALL abbreviations, table 1 is missing E, CAZ, IMP, table 2 is missing RTU (table 3 should potentially include cell abbreviations if required)

4.      Figure 1 is unacceptable in it’s present form, figures should never be cut across 2 pages (unless to prevent loss of clarity in large datasets). The first Bland-Altman plot should be labelled A and placed to the left of the counts graph (which should be labelled B), with the table labelled C.

5.      A “no bacteria” control plot should be added to Figure 2, to show that the events detected are bacteria (and/or bacterial components) and not just general background noise.

6.      Figure 3 E. coli K12 tested (top row) shows 3 events/μL at 0.078% Sterisol, but then at 0.078% there are 238 events/μL. This makes no sense.

7.      In the text the concentration is described as MBC/2, but in the figure labelled as 1/2 MBC – this needs to be consistent.

8.      Figure 7 legend describes “(B) Syto24 and CTC or (C) Syto9” yet the figure does not have the appropriate B or C labels.

9.      As mentioned in point 5 above, a no bacteria control needs to be included. In the discussion (lines 393-401) the authors describe this method being used to rapidly identify detergents used to kill bacteria, but each bacteria requires a different gating strategy and there is no evidence presented that the bacteria gates are specific (due to lack of no bacteria control).

Minor Comments:

1.      Page 2, line 66, mem-brane should have hyphen removed (membrane).

2.      Figure 1, counts/mL figure should have scale reduced – there is unnecessary blank space at the top

3.      The “patents” (section 6) and “Appendix” (A and B sections) have generic text. These should be removed (or completed specific to the article).

Comments on the Quality of English Language

The English could be minorly improved (but is sufficient for publication)

Round 2

Reviewer 1 Report (Previous Reviewer 1)

Comments and Suggestions for Authors

The authors have made significant improvement to the work. The table with the highlights of the novelty of the work, compared with previous studies, should be added to the manuscript.

Author Response

Reviewer' Comment to the Authors:

The authors have made significant improvements to the work. The table with the highlights of the novelty of the work, compared with previous studies, should be added to the manuscript.

Author response:  

We have included the Table in the Introduction section with highlights of the novelty of our method in comparison with previous studies. We thank the reviewer for helping us improve our submission.

Reviewer 2 Report (New Reviewer)

Comments and Suggestions for Authors

The authors have incorporated all changes, in some places extensively so, to address all previous concerns with the manuscript.

Author Response

We thank the reviewer for helping us improve our submission.

This manuscript is a resubmission of an earlier submission. The following is a list of the peer review reports and author responses from that submission.

Round 1

Reviewer 1 Report

Comments and Suggestions for Authors

Despite the current manuscript being an interesting study on the assessment of quaternary ammonium compounds disinfectants’ bactericidal activity by flow cytometry, this research is not by any means novel/new, as seen in already published literature that dates back to 1996 https://pubmed.ncbi.nlm.nih.gov/8896352/, and many other similar studies, such as:

https://www.frontiersin.org/journals/microbiology/articles/10.3389/fmicb.2022.1023326/full

https://bmcmicrobiol.biomedcentral.com/articles/10.1186/s12866-020-01818-3

https://archive.hshsl.umaryland.edu/handle/10713/1546

https://www.sciencedirect.com/science/article/pii/S0223523418310602

https://www.mdpi.com/1420-3049/24/3/532

https://www.frontiersin.org/journals/microbiology/articles/10.3389/fmicb.2017.00112/full

Given this complete lack of novelty, I do not recommend the publication of this research in a high impact Q1 journal such as IJMS.

Reviewer 2 Report

Comments and Suggestions for Authors

The manuscript entitled “A new approach based on flow cytometry for rapid testing of quaternary ammonium compounds disinfectants’ bactericidal activity” is well written and well-structured by the Authors who denote that they possess considerable scientific rigor in their approach to research.

I just need to make some small points:

-            Making the results section precede the materials and methods section makes it more difficult to interpret and read the entire manuscript.

-            The figures are absolutely essential, but in some cases almost uninterpretable because of their excessively small size. I refer in particular to Figures 2, 7 and 8. It would be desirable if the Authors could produce a more intelligible version.

-            Although there are illustrations in the supplementary material those that the Authors have appropriately decided to include in the main manuscript must be legible.

-            Specific contributions of individual Authors are not indicated in the dedicated section at the end of the main manuscript.

In other respects, I believe that the authors have done a high quality job that needs only minor but not substantial formal adjustments.

Round 2

Reviewer 1 Report

Comments and Suggestions for Authors

Regarding the author’s reply, label-free cytometry for bacteria assessment is not novel either, as already published in:

https://pubs.acs.org/doi/10.1021/acs.analchem.9b01869

https://www.ncbi.nlm.nih.gov/pmc/articles/PMC3754729/

https://europepmc.org/article/med/22243282

These studies also used the bacteria’s autofluorescence.

Additionally, other label-free flow cytometry methods for bacteria detection have also been developed already:

https://www.mdpi.com/1424-8220/18/10/3496

https://www.researchgate.net/publication/374829508_Label-free_multidimensional_bacterial_characterization_with_ultrawide_detectable_concentration_by_microfluidic_impedance_cytometry

Thus, the novelty of the developed assays is still lacking.

Author Response

We thank the reviewer for taking the time to analyze our research. With regards to the lack of novelty of our research, the examples provided by the reviewer, however does not support the fact that our research does not bring new knowledge. 

https://pubs.acs.org/doi/10.1021/acs.analchem.9b01869 - the authors used autofluorescence from flavin to detect and quantify bacteria in juice. The FL1 (green channel) was used to quantify different bacteria.

https://www.ncbi.nlm.nih.gov/pmc/articles/PMC3754729/ and https://europepmc.org/article/med/22243282-detection of bacterial autofluorescence.

Regarding the following examples:

https://www.mdpi.com/1424-8220/18/10/3496

https://www.researchgate.net/publication/374829508_Label-free_multidimensional_bacterial_characterization_with_ultrawide_detectable_concentration_by_microfluidic_impedance_cytometry

These methods used impedance cytometry not flow cytometry.

Our proposed protocol used forward and side scatter signal measurements for fast and accurate quantification of viable bacteria after exposure to QACs. The developed protocol was not based on fluorescence measurements and did not used impedance cytometry.

We will resubmit our paper highlighting better the originality of our research.

Thank you very much

Kind regards